

# Microplastic from beach sediment to tissue: a case study on burrowing crab *Dotilla blanfordi*

Hiralba Zala[1,*], Vasantkumar Rabari[1,*], Krupal Patel[1], Heris Patel[1], Virendra Kumar Yadav[1], Ashish Patel[1], Dipak Kumar Sahoo[2] and Jigneshkumar Trivedi[1]

[1] Department of Life Sciences, Hemchandracharya North Gujarat University, Patan, Gujarat, India
[2] Department of Veterinary Clinical Sciences, Iowa State University, Ames, Iowa, United States
* These authors contributed equally to this work.

Corresponding authors
Dipak Kumar Sahoo,
dsahoo@iastate.edu
Jigneshkumar Trivedi,
jntrivedi26@yahoo.co.in

## ABSTRACT

**Background:** Microplastics (MPs) are pervasive pollutants in the marine environment, exhibiting persistence in coastal sediment over extended periods. However, the mechanism of their uptake by marine organisms and distribution in habitat is less understood. The objective of the present study was to investigate the presence of MP contamination in burrow sediment, feeding pellets, and tissue of *Dotilla blanfordi* in the Gulf of Kachchh, Gujarat State.

**Methods:** A total of 500 g of burrow sediment, 100 g of feeding pellets, and body tissue of 10 resident *D. blanfordi* were pooled as one replica. Such seven replicas from each site were analyzed for MP extraction from three sites, including Asharmata, Mandvi, and Serena, located in the Gulf of Kachchh. The standard protocol was used during the analysis of the collected samples in order to isolate MPs.

**Results:** The abundance of MP was found higher in burrow sediment, feeding pellets and tissue of *D. blanfordi* at study site Mandvi, followed by Serena and Asharmata. The abundance of MP was found higher in *D. blanfordi* tissue, followed by burrow sediment and feeding pellet. A significant variation was observed in MP abundance among burrow sediment, feeding pellets, and tissue. MPs with various shapes (fiber, film, and fragment), sizes (1–2, 2–3, 3–4, and 4–5 mm), and colors (blue, green, black, pink, purple, red transparent) were recorded from all the study sites. Polyurethane and polyvinyl chloride were recognized as the chemical profile of the extracted MPs. The current investigation revealed greater accumulation of MPs in *D. blanfordi*'s tissues compared to sediment and pellets, suggesting a risk of MP contamination in marine benthic fauna with a greater rate of bioaccumulation. *D. blanfordi* plays a significant role as a structuring agent for MP distribution in the intertidal flat through burrowing activity.

## INTRODUCTION

Plastics are synthetic or semi-synthetic organic polymers known for their lightweight nature, durability, affordability, longevity, and resistance to corrosion (*Derraik, 2002*).

The prevalence of plastic debris in the ocean was a consequence of exponential usage and a lack of waste management practices (*Browne, Galloway & Thompson, 2010*). In the early 19th century, plastic pollution had drawn minimal attention to the scientific community (*Carpenter et al., 1972*). However, in today's scenario, no habitat on earth has escaped plastic pollution (*Barnes et al., 2009*; *Rabari et al., 2024*). Due to photo- and thermo-degradation, accumulated plastic debris fragmented into pieces of plastic (*Rabari et al., 2022*). Plastic particles having 1 µm to 5 mm in size are called microplastics (MP) (*Lippiatt, Opfer & Arthur, 2013*; *Hartmann et al., 2019*). Two known types of MPs are primary and secondary MPs. Primary are deliberately produced for use in cosmetics and as carriers for drugs (*Rabari et al., 2023a*). Secondary MPs arise from the breakdown, photo-thermal degradation, and chemical deposition of larger plastic debris over time (*Thompson et al., 2004*; *Andrady, 2011*). MPs can be found in various forms, such as fibers, pellets, films, foam, and fragments (*Daniel, Ashraf & Thomas, 2020*).

The widespread distribution of MP in oceanic water and coastal sediment can lead to the false ingestion of MP by biota (*Rabari et al., 2023a*: *Patel et al., 2024*; *Joshi et al., 2024*). MP has been recorded in corals (*Huang et al., 2021*), jellyfish (*Rapp et al., 2021*), sea cucumber (*Mohsen et al., 2021*), shrimp (*Daniel, Ashraf & Thomas, 2020*), oysters (*Teng et al., 2019*), calm (*Serra & Colomer, 2023*), crab (*Wang et al., 2021*), fish (*Wang, Ge & Yu, 2020*), and sea birds (*Hamilton et al., 2021*). Furthermore, there have been observations of trophic transfer resulting from the bioaccumulation and magnification of MPs. For instance, *Farrell & Nelson (2013)* reported the presence of MPs in the shore crab *Carcinus maenas*, which had ingested *Mytilus edulis*. Similarly, *Crooks, Parker & Pernetta (2019)* noted the retention of MPs in the gut of the velvet crab *Necora puber* after consuming MP-contaminated mussels, indicating a potential pathway for the transfer of MPs through the food web. The accumulation of MPs in various body organs, such as the gut, gills, intestines, liver, and muscles, can lead to toxic effects on the organism's health (*Pirsaheb, Hossini & Makhdoumi, 2020*). MP contamination can cause detrimental effects on the organism's body, including starvation, impaired feeding capacity, reduced drowning, and reproductive fitness (*Kach & Ward, 2008*; *Yu et al., 2018*). At the cellular level, MP can cause cell damage, pathological stress, oxidative stress, genotoxicity, neurotransmission dysfunction, and ultimately impaired immunity (*Wright, Thompson & Galloway, 2013*; *Avio et al., 2015*; *Pitt et al., 2018*). Exposure of crab *E. sinensis* to MPs was found to elevate the expression of the p38 gene, potentially leading to the induction of oxidative stress (*Yu et al., 2018*). Moreover, severe hepatic damage and impaired neural mechanisms were observed in the crab *Charybdis japonica* due to exposure to MP greater than 3 mg/g (*Wang et al., 2021*).

Extensive studies were conducted on various groups of crustaceans inhabiting the coastal areas of Gujarat State to understand their diversity (*Trivedi & Vachhrajani, 2013*, *2017*; *Trivedi, Soni & Vachhrajani, 2015*; *Trivedi, Osawa & Vachhrajani, 2016*; *Trivedi, Trivedi & Vachharajani, 2017*; *Trivedi, Campos & Vachhrajani, 2018*; *Trivedi et al., 2015a*, *2015b*, *2018*, *2020*; *Patel et al., 2022*; *Thacker et al., 2023*) and ecology (*Trivedi, 2016*; *Trivedi & Vachhrajani, 2012*; *Trivedi & Vachhrajani, 2014a*, *2014b*). Among these, crabs are widely distributed omnivores that are sensitive to environmental changes and

frequently employed as pollution indicators (*Aheto et al., 2011*; *Jonah et al., 2015*; *Doshi et al., 2024*). *Dotilla blanfordi* is a common burrowing crab species found in tropical areas that remains buried during high tide and emerges during low tide (*Upadhyay et al., 2022*). Their feeding mechanisms involve the filtering of organic detritus from burrow sediment and subsequently converts leftover sediment into feeding pellets (*Hartnoll, 1973*; *Gherardi & Russo, 2001*; Fig. 1). Burrowing and feeding patterns play a key role in nutrition recycling and the dynamic enhancement of soil aeration (*Kristensen et al., 2008*). However, the feeding mechanism of *D. blanfordi* makes them more prone to ingesting MP along with organic detritus and can alter the concentration of MPs in sediments. It is the most common burrowing crab species found in the muddy shores of Gujarat State, and some ecological aspects of this species have been studied well (*Mitra, Trivedi & Mendoza, 2020*; *Gajjar, Patel & Trivedi, 2021*; *Joshi, Patel & Trivedi, 2022*; *Desai et al., 2022*), but the assessment of MP contamination has not been studied for this species. Therefore, the current investigation pointed to examining the contamination of MPs in the burrow sediment, feeding pellets, and tissues of *D. blanfordi*.

## MATERIALS AND METHODS

### Study area

Gujarat state has the longest coastline (~1,600 km) in India, supporting rich floral and faunal diversity in a variety of habitats. The present study was conducted at the Kachchh Gulf from November 2022 to December 2022. A total of three beaches (Asharmata, Serena, and Mandvi) were selected based on anthropogenic pressure (*Gül & Griffen, 2018*; *Rabari et al., 2022*; *Shinde et al., 2024*; Fig. 2). The visitors were counted using binoculars for 2 h over a one-kilometer linear transect. Moreover, possible sources of plastic, beach cleaning activities, vehicles, and tire marks were taken into consideration for the classification of beaches. In terms of anthropogenic pressure, the study site Asharmata was considered as a low-impact site (<30 visitors/h), followed by Serena as a moderately impacted site (30–50 visitors/h), and Mandvi as a highly impacted site (>50 visitors/h) (*Rabari et al., 2022*). The possible plastic input in the study site Asharmata can be tourist attractions, plastic wrappers, and bottles, bags. The possible input of plastic in Serena Beach, Kutch, is likely from nearby coastal activities, tourist littering, and fishing industry waste. The possible input of plastic in Mandvi Beach, Kutch, is likely from tourist activities, local waste disposal practices, and fishing-related debris.

### Methodology for sample collection

The sample collection procedure was carried out after 2 h of the commencement of low tide (less than 0.5 m) from November to December 2022 because maximum feeding activity by *D. blanfordi* is finished in the initial 2 h (*Desai et al., 2022*). A total of seven replicas of samples were examined for each site during the study. For the preparation of each replica, the following method was used (*Capparelli et al., 2022*). A total of 10 burrows of *D. blanfordi* were randomly selected to prepare one replica. From each selected burrow, resident *D. blanfordi*, burrow sediment (~using a stainless-steel core, reaching depths of up to 30 cm), and all feeding pellets (using a stainless-steel scalper) were collected. The burrow
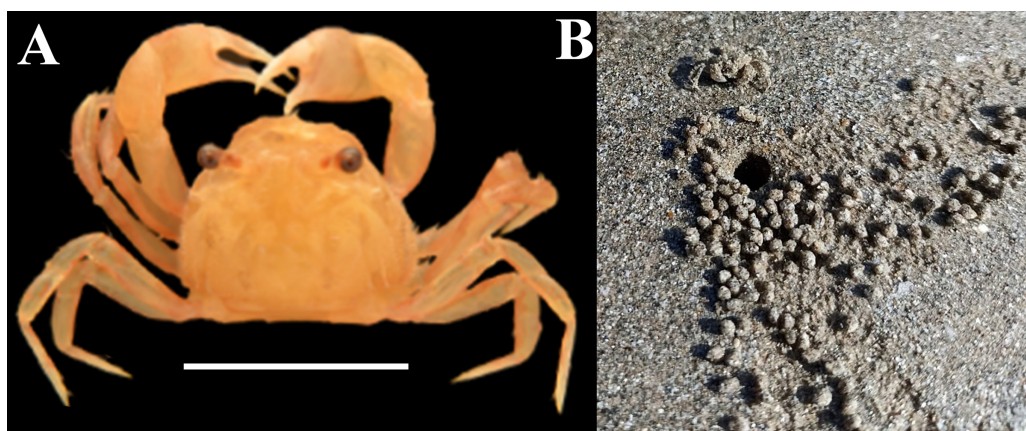

**Figure 1 (A) Representative photograph of burrowing crab *D. blanfordi*, (B) burrowing behavior, and representative photograph of feeding pellets.** Image prepared in Photoshop CS6.

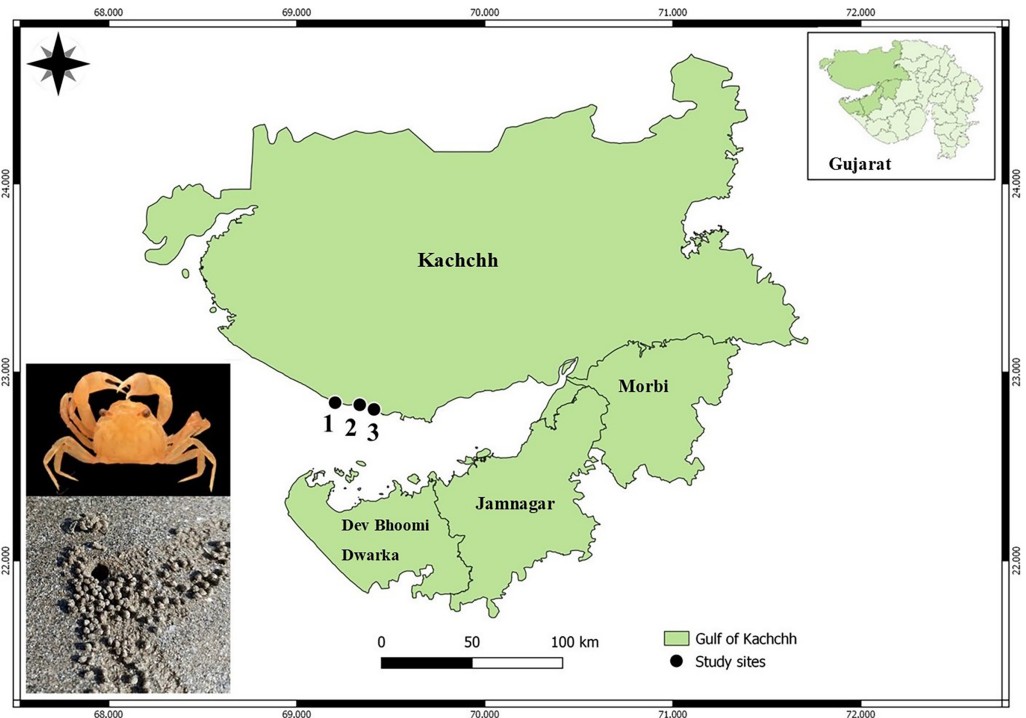

**Figure 2 Map of the sampling area highlighting geographic locations: (1) Asharmata, (2) Mandvi, and (3) Serena.** Map prepared using QGIS 3.14 software.

sediment collected from each burrow is combined in a steel crate at the field site. Similarly, the collected feeding pellets from each burrow were combined in a separate steel crate. Later, 500 g of burrow sediment and 100 g of feeding pellets were taken as one replica for further analysis. The resident, *D. blanfordi*, was captured using the hand-picking method, placed in a steel container, and brought to the laboratory in ice boxes. The body tissues of 10 resident *D. blanfordi* were used for each replica.

## MPs extraction from burrow sediment and feeding pellet

From each replica, the collected samples of 500 g of burrow sediments and 100 g of pellets were subjected to drying at 60 °C in a hot-air oven for a period of 48 h. From each replica, three sub-replicates of 20 g were prepared and passed through stainless-steel sieves of 2, 1, 0.5, and 0.25 mm in size. The sediments captured in each sieve were weighed and then transferred to the beaker. The organic matter present in the sediment was digested using 30% hydrogen peroxide. Subsequently, a supersaturated solution of sodium chloride (360 g/L) was added to the beaker to facilitate the flotation of the MPs, utilizing the principle of the density gradient. The mixture was agitated using a glass rod and kept at room temperature to settle the MP particles. The solution was passed through Ashless Whatman filter paper (Grade No. 41, pore size: 20 μm), and the filter papers were subsequently left to dry at room temperature.

## MPs extraction from *D. blanfordi*'s tissue

The collected *D. blanfordi* specimens in each replica were cleaned using Milli-Q water to remove surface contaminants adhered to the *D. blanfordi* body. The *D. blanfordi*'s body weight was measured using a digital weighing balance. Each *D. blanfordi* was dissected using a stainless-steel dissection kit. The tissue of 10 pooled *D. blanfordi* was churned with the mortar pastel and placed in a beaker. The chemical digestion of organic tissue was performed using a 10% potassium hydroxide solution at a temperature of 60 °C for a duration of 48 h (*Li et al., 2015*). After the digestion of organic tissue, a highly concentrated NaCl solution was employed to facilitate the flotation of the MPs. The solution containing the floated particles was filtered using Ashless-Whatman filter paper and left to dry at room temperature.

## Quantification and chemical identification of MPs

MPs were counted and quantified in order to morphometric characters (shape, size, and color). The representative photographs of each MP's shape were captured under a stereomicroscope. ATR-FTIR was used to identify the chemical profile of MPs at the CIMF laboratory, HNGU, Patan. Out of the total isolated MPs, 10% were selected from representative shapes of MPs for identification of polymer composition. The acquired spectra were compared with established polymer libraries (FLOPP and FLOPP-e, consisting of 762 spectra) (*De Frond, Rubinovitz & Rochman, 2021*). Above the 70% match, particles were considered as MPs.

## Contamination control

Collected samples of burrow sediments, feeding pellets, and *D. blanfordi* were covered properly with aluminium foil to prevent environmental contamination. Prior to analysis, *D. blanfordi* specimens were pre-cleaned in Milli-Q water to eradicate any adhering contaminants. Before use, the metal tray and stainless-steel utensils were cleaned with Milli-Q water during the dissection process. Sample analysis and the isolation of MPs were conducted in a secluded area with minimal human activity. Moreover, three blanks with

water run simultaneously from the digestion to filtration steps. No MPs were recorded in blanks.

## Data analysis

To check the concentration of MPs, a mean and standard deviation of MP contamination (MPs/g) in burrow sediment, feeding pellet, and tissue were calculated. The proportions of shape, size, and color were computed as percentages. To assess the distribution of the data, a Shapiro-Wilk test was incorporated. Due to the non-normal distribution of the data (Figs. S1–S3), a non-parametric test was performed. To examine the variation of MP contamination among burrow sediment, feeding pellets, and tissue at each site, a Kruskal-Wallis test was incorporated. Additionally, a Kruskal-Wallis test was employed to comprehend the disparity of MP in burrow sediment, feeding pellets, and tissue across the various study sites. The data analysis was conducted using R Studio and MS Excel (Microsoft, Redmond, WA, USA).

# RESULTS AND DISCUSSION

## Concentration of MPs

In the current investigation, MP in burrow sediment, pellet, and tissue of burrowing crab *D. blanfordi* was investigated at three sandy beaches of the Gulf of Kachchh, Gujarat State. The presence of 100% MP was detected in burrow sediment, feeding pellets, and *D. blanfordi*'s tissue collected from all the study sites. At the Asharmata study site, 417, 223, and 105 MPs were found in burrow sediments, feeding pellets, and *D. blanfordi* tissue, respectively. A total of 523, 338, and 221 MPs were found in burrow sediments, feeding pellets, and *D. blanfordi*'s tissue collected from study site Mandvi. A total of 498, 283, and 114 MPs were recorded in burrow sediments, feeding pellets, and *D. blanfordi*'s tissue collected from study site Serena. The abundance of MP was found higher in study site Mandvi (1.25 ± 0.51 MPs/g in burrow sediment, 0.80 ± 0.21 MPs/g in feeding pellet, and 3.16 ± 1.44 MPs/g in tissue), followed by Serena (1.19 ± 0.68 MPs/g in burrow sediment, 0.67 ± 0.14 MPs/g in feeding pellet, and 1.63 ± 0.30 MPs/g in tissue), and Asharmata (0.99 ± 0.17 MPs/g in burrow sediment, 0.53 ± 0.07 MPs/g in feeding pellet, and 1.5 ± 0.86 MPs/g in tissue) (Fig. 3). MPs contamination was recorded among different classified study sites as follows: highly impacted, moderately impacted, and low-impacted sites. It was found that the highly impacted site of Mandvi has shown higher contamination of MPs in burrow sediment, feeding pellets, and tissue. The higher abundance of MPs in Mandvi Beach possibly relates to tourism, beach development activities, fishing, sewage discharge, and industrial pollution. The moderately impacted study site Serena has shown moderate MP contamination in burrow sediment, feeding pellets, and tissue. The low-impacted sites of Asharmata have shown low MP contamination in burrow sediment, feeding pellets, and tissue. Similarly, *Rabari et al. (2022)* recorded variation in MP accumulation between highly impacted and low-impacted sites. The disparity in MP contamination among the various study sites appeared to be influenced by varying levels of anthropogenic activities observed on beaches (*Dowarah & Devipriya, 2019*; *Patchaiyappan et al., 2021*).

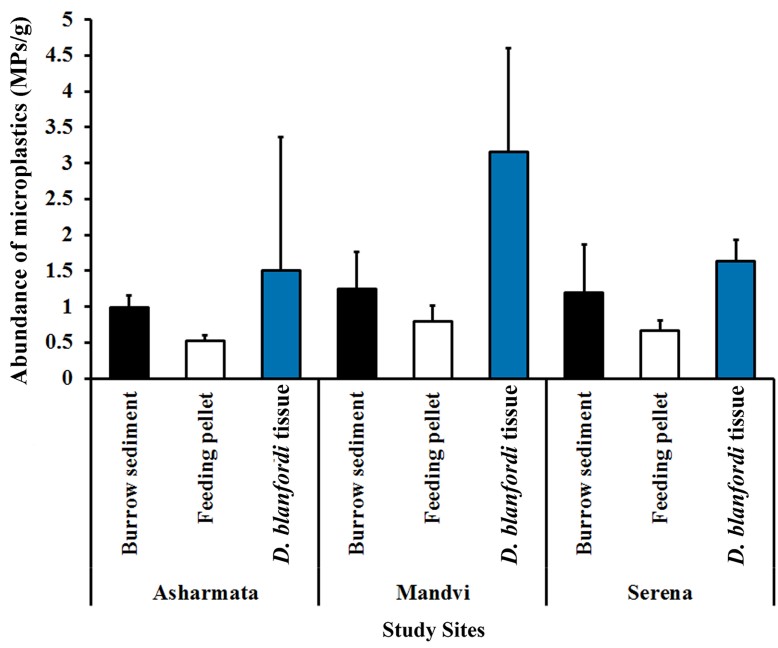

**Figure 3 Abundance of MP contamination in burrow sediment, feeding pellet, and tissue *D. blanfordi* collected from the Gulf of Kachchh.**

**Table 1 *Post-hoc* test results demonstrating variation in MPs between burrow sediment, feeding pellet and tissue.**

|  | Tissue | Feeding pellet |
|---|---|---|
| **Study site asharmata** | | |
| Feeding pellet | 0.4017 | – |
| Burrow sediment | 0.4017 | 0.0064 |
| **Study site Mandvi** | | |
| Feeding pellet | 0.0017 | – |
| Burrow sediment | 0.0082 | 0.0474 |
| **Study site Serena** | | |
| Feeding pellet | 0.0063 | – |
| Burrow sediment | 0.1095 | 0.1095 |

The abundance of MPs was recorded higher in tissue, followed by burrow sediment and feeding pellet (Fig. 3). Moreover, the abundance of MP contamination varied significantly between burrow sediment, feeding pellet, and tissue at study site Asharmata (H ($\chi^2$) = 6.91, $p$ = 0.03, df = 2), at study site Mandvi (H ($\chi^2$) = 14.05, $p$ = 0.0008, df = 2), and at study site Serena (H ($\chi^2$) = 11.75, $p$ = 0.002, df = 2). The *post-hoc* test results highlighted the variation in MP abundance within the group of burrow sediment, feeding pellet, and tissue (Table 1). A comparison of MP in sediment and tissue is presented in Fig. 4, Tables 2 and 3. Burrowing crabs are responsible for turning over a large volume of sediment in the uppermost layer of the intertidal region (*Litulo, 2005*). The burrowing mechanism of

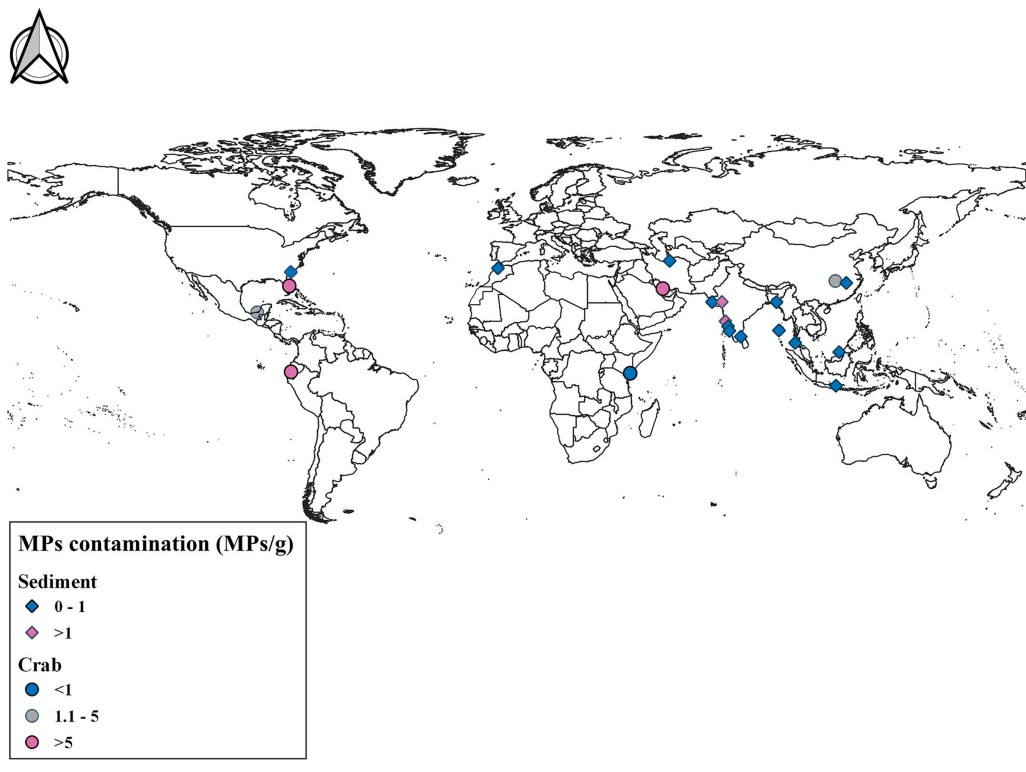

**Figure 4 Comparison of MPs contamination in sediment and crab species across world.** Map prepared using QGIS 3.14 software.

**Table 2 Comparison of average abundance of MPs contamination in coastal sediment globally.**

| Location | Microplastic contamination (MPs/g) | Reference |
|---|---|---|
| Gulf of Kachchh, Gujarat, India | 1.14 ± 0.45 MPs/g in burrow sediment | Present study |
| Changjiang Estuary, China | 0.121 | *Peng et al. (2017)* |
| Karnaphuli River Estuary, Bangladesh | 0.040895 | *Rakib et al. (2022)* |
| Baram River estuary, Borneo Island, Malaysia | 0.6944 | *Choong et al. (2021)* |
| Jagir Estuary, Surabaya City, Indonesia | 0.59 | *Firdaus, Trihadiningrum & Lestari (2020)* |
| Qarasu estuary in Gorgan Bay, south-east of Caspian Sea, Iran | 0.25 | *Gholizadeh & Cera (2022)* |
| Sebou Estuary and Atlantic Coast, Morocco | 0.155 | *Haddout et al. (2022)* |
| Phuket province, Thailand | 0.234 | *Jiwarungrueangkul et al. (2021)* |
| Daniel Island, Charleston Harbor estuary, South Carolina, USA | 0.0336 | *Leads & Weinstein (2019)* |
| Vellar Estuary, Tamil Nadu | 0.0681 | *Nithin, Sundaramanickam & Sathish (2022)* |
| Andaman and Nicobar Islands | 0.41435 | *Patchaiyappan et al. (2020)* |
| Estuarine system from Central West coast, Goa | 6.093 | *Gupta et al. (2021)* |
| Karnataka | 0.633 | *Yaranal, Subbiah & Mohanty (2021)* |
| Kavvayi and Kumbla Estuaries, Kerala | 0.098 | *Padmachandran et al. (2023)* |
| Sandy beaches of Gujarat | 0.0137 | *Rabari et al. (2022)* |
| Gulf Of Khambhat | 1.565 | *Rabari et al. (2023a, 2023b, 2023c)* |

**Table 3 Comparison of abundance of MP contamination across various crab species worldwide.**

| Location | Species | Microplastic contamination (MPs/g) | Reference |
|---|---|---|---|
| Gulf of Kachchh, Gujarat, India | *Dotilla blanfordi* | 2.09 ± 0.86 MPs/g in tissue and 0.66 ± 0.14 MPs/g in feeding pellet | Present study |
| Isla Santay, Ecuador | *Leptuca festae* | 18.69 | *Villegas, Cabrera & Capparelli (2021)* |
| Isla Santay, Ecuador | *Minuca ecuadoriensis* | 11.715 | *Villegas, Cabrera & Capparelli (2021)* |
| Kenyan Coast | *Tubuca dussumieri* | 0.685 | *Awuor, Muthumbi & Robertson-Andersson (2020)* |
| Kenyan Coast | *Cranuca inversa* | 0.425 | *Awuor, Muthumbi & Robertson-Andersson (2020)* |
| Kenyan Coast | *Gelasimus vocans* | 0.79 | *Awuor, Muthumbi & Robertson-Andersson (2020)* |
| Persian Gulf | *Portunus armatus* | 0.5595 | *Akhbarizadeh, Moore & Keshavarzi (2019)* |
| Indian River Lagoon system | *Panopeus herbstii* | 297.74 | *Waite, Donnelly & Walters (2018)* |
| Chongming Island, Yangtze Estuary | *Chiromantes dehaani* | 1.9 | *Wu et al. (2020)* |
| Isla del Carmen | *Minuca rapax* | 1.3 | *Capparelli et al. (2022)* |

*D. blanfordi* not only modulates the distribution of MPs on the surface but also alters within the burrow. The MP contamination was found to be higher in tissue, followed by burrow sediment and feeding pellet. The possible explanation for the higher MPs in tissue and burrow sediment compared to feeding pellets can be easily understood by considering the feeding pattern of *D. blanfordi*. They use their water-filled mouths to strain and separate organic detritus from the burrow sediment, subsequently forming feeding pellets with the leftover sediment in their mouths (*Hartnoll, 1973*; *Gherardi & Russo, 2001*). Filtering of organic detritus along with MP particles can lead to the event of false ingestion of MP particles by *D. blanfordi*. The less MP contamination in the feeding pellets may be attributed to the efficient filtering mechanism, which effectively removes MPs from the sediment during the filtration process. It is quite imperative to note that the dynamic nature of MP distribution by bioengineers is species-specific. *Capparelli et al. (2022)* have checked the alteration in MP distribution on Mexican beaches through the bioturbation of *Minuca rapax*.

## Physical and chemical properties of extracted MPs

The physical (shape, color, and size) and chemical properties of the extracted MPs were assessed. In terms of shape of MPs, fibers were recorded pre-dominantly, followed by fragments and films in burrow sediment, feeding pellets, and tissue in all the study sites (Fig. 5A). Photographs representing each shape of the isolated MPs were captured (Fig. 5B). Similarly, fibers were recorded dominantly in sediment collected from the same study region, the Gulf of Kachchh (*Rabari et al., 2022*), and in crab *Carcinus aestuarii*

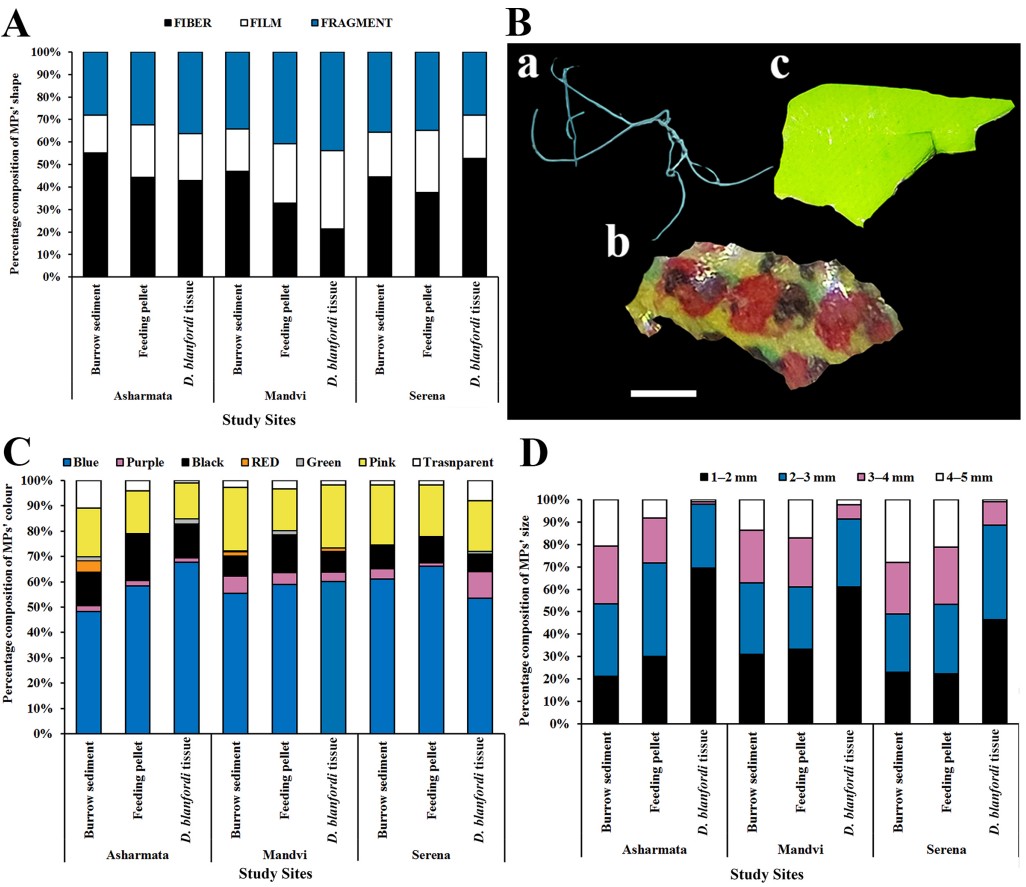

**Figure 5** Physical properties of MPs (A) shapes of MPs found in burrow sediment, feeding pellets, and tissue collected from the Gulf of Kachchh, (B) photographs of MPs shapes a-fiber, b-film, c-fragment (scale represents 1 mm), (the figure was prepared using Photoshop CS6 software) (C) colors of MPs, and (D) size classes of MPs.

(*Piarulli et al., 2019*), *M. rapax* (*Capparelli et al., 2022*), and *Leptuca festae* (*Villegas, Cabrera & Capparelli, 2021*). Conversely, *Stasolla, Innocenti & Galil (2015)* observed a prevalence of fragments in *Charybdis longicollis*, while *Akhbarizadeh, Moore & Keshavarzi (2019)* noted a similar trend in *Portunus armatus*. Fibers in the marine environment could originate from sources like fishing nets or wastewater discharge (*Feng et al., 2019*). It was found that smaller plastic particles, displaying a range of shapes, result from the breakdown of larger plastic debris through fragmentation and photo-thermo degradation (*Sathish, Jeyasanta & Patterson, 2019*). In terms of colors of MPs, blue, pink, and black-colored MPs were found dominantly, followed by transparent, red, purple, and green in burrow sediment, feeding pellet, and tissue in all the sampling locations (Fig. 5C). Similarly, black and blue-colored MPs were recorded dominantly in the sediment of Gujarat coast (*Rabari et al., 2022*) and the crabs *Ocypode quadrata* (*Costa et al., 2019*) and *Portunus pelagicus* (*Kleawkla, 2019*). The study conducted by *Prusty et al. (2023)* suggests that fishing nets of black and blue color could potentially be an origin of addressed colored MPs (*Prusty et al., 2023*). Additionally, it was found that marine organisms may

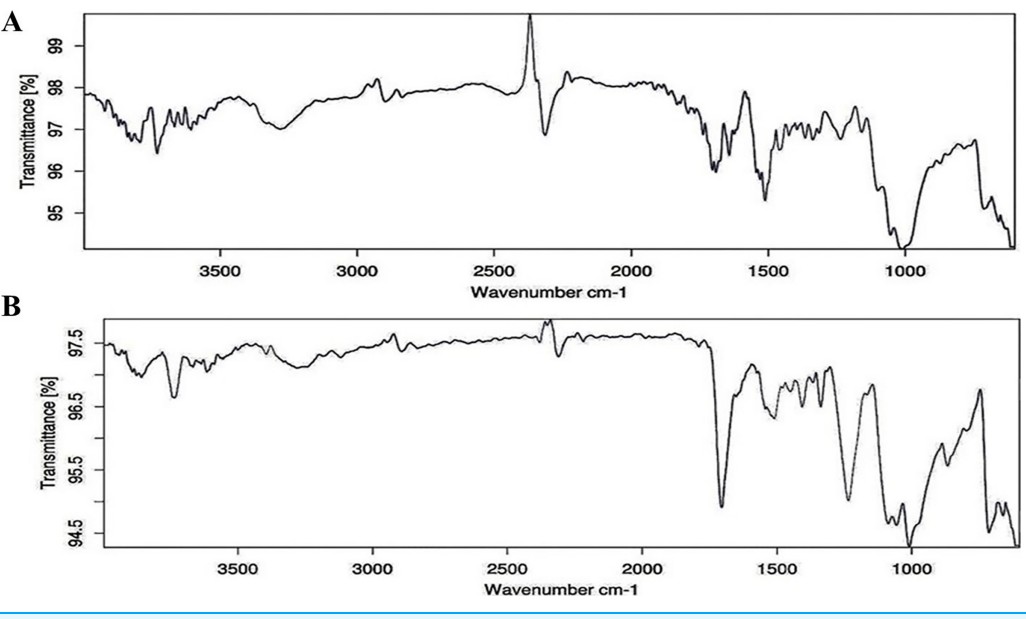

**Figure 6 ATR-FTIR spectrum of extracted MPs (A) PU, and (B) PVC.**

mistakenly consume black and blue-colored MPs because they resemble their natural prey, resulting in deceptive ingestion (*Wright, Thompson & Galloway, 2013*).

In terms of size classification, it was found that MPs within the 1–2 mm range were the most prevalent, followed by those in the 2–3, 3–4, and 4–5 mm categories (Fig. 5D). Similarly, *Rabari et al. (2022, 2023a)* have recorded the dominance of 1–2 mm-sized MPs in beach sediment of Gujarat State. The feeding pattern of the organism can impact the prevalence of different sizes of MPs (*D'Costa, 2022*). *Rabari et al. (2023b)* highlighted that the production of smaller particles stems from the breakdown of larger plastic waste *via* photo-thermo degradation. MPs of smaller sizes have a higher tendency to absorb adhesive pollutants owing to their increased surface area (*Robin et al., 2020*). The ATR-FTIR was used to know the chemical properties of the extracted MPs. Comparing the acquired spectra with established plastic libraries indicated that the extracted MPs were composed of polyurethane (PU) and polyvinyl chloride (PVC) (Fig. 6). The identification of the chemical composition of isolated MPs can aid in discerning the source of these particles (*Rabari et al., 2023c*). The PU can potentially be used in marine equipment, medical devices, sealants, and adhesives (*Zia, Bhatti & Bhatti, 2007*). The sources of PVC can be cable ducting, telecom wiring and cables, flooring, window and door profiles, waste effluent discharge, pipes, and fittings (*Lewandowski & Skórczewska, 2022*). The present study highlighted alteration in MP distribution by the foraging behavior of burrowing crab *D. blanfordi* of the Gulf of Kachchh, Gujarat State. The presence of MPs in marine organisms can lead to detrimental consequences for species, including issues such as reduced food intake, inhibited growth, reproductive abnormalities, blockages in the gastrointestinal tract, and even increased mortality (*Rabari et al., 2022*; *Song et al., 2023*; *Oza et al., 2024*).

## CONCLUSION

Prior to reaching a conclusion, it is crucial to address the limitations of the study. The study limits the appropriate comparison between sediment unaffected by crabs and sediment having crab burrows, which may provide interesting insight into the effects of feeding on MP distribution. The current investigation evaluated the contamination of MPs in burrow sediment, feeding pellets, and organic tissue of the burrowing crab *D. blanfordi* in the Gulf of Kachchh, Gujarat State. At the study sites, Mandvi exhibited a higher recorded average abundance of MP contamination, followed by Serena and Asharmata. Tissue showed a higher abundance of MP contamination, followed by burrow sediment and feeding pellet. A significant variation was recorded between burrow sediment, feeding pellet, and tissue at Asharmata, Mandvi, and Serena. Fibers were recorded dominantly, followed by fragments and films in burrow sediment, feeding pellets, and tissue in all the sampling locations. Predominantly, MPs in blue, pink, and black colors, with sizes ranging from 1 to 2 mm, were recorded. PU and PVC were identified as the chemical compositions of the isolated MPs. Fishing nets, marine equipment, sealants, adhesives, and tourism can be the possible input of MPs in the ocean of Gujarat State. The study highlighted an alteration in MP distribution through the burrowing behavior of *D. blanfordi*. The findings underscore the urgent necessity for effective management of plastic waste in the marine ecosystem of Gujarat State, India.

## ACKNOWLEDGEMENTS

The authors also acknowledge the Department of Life Sciences, Hemchandracharya North Gujarat University, Patan, Gujarat, India, for their provision of laboratory facilities and additional technical support.

### Funding

The authors received no funding for this work.

### Competing Interests

The authors declare that they have no competing interests.

### Author Contributions

- Hiralba Zala conceived and designed the experiments, performed the experiments, prepared figures and/or tables, authored or reviewed drafts of the article, and approved the final draft.
- Vasantkumar Rabari conceived and designed the experiments, performed the experiments, analyzed the data, prepared figures and/or tables, authored or reviewed drafts of the article, and approved the final draft.
- Krupal Patel conceived and designed the experiments, performed the experiments, prepared figures and/or tables, and approved the final draft.

- Heris Patel conceived and designed the experiments, performed the experiments, prepared figures and/or tables, and approved the final draft.
- Virendra Kumar Yadav conceived and designed the experiments, performed the experiments, analyzed the data, prepared figures and/or tables, authored or reviewed drafts of the article, and approved the final draft.
- Ashish Patel conceived and designed the experiments, analyzed the data, authored or reviewed drafts of the article, and approved the final draft.
- Dipak Kumar Sahoo conceived and designed the experiments, analyzed the data, authored or reviewed drafts of the article, and approved the final draft.
- Jigneshkumar Trivedi conceived and designed the experiments, analyzed the data, authored or reviewed drafts of the article, and approved the final draft.

### Data Availability

The raw data are available in the Supplemental File.

### Supplemental Information

Supplemental information for this article can be found online at http://dx.doi.org/10.7717/peerj.17738#supplemental-information.

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
