# Peer review of "Microplastic from beach sediment to tissue: a case study on burrowing crab Dotilla blanfordi"

_PeerJ, doi:10.7717/peerj.17738_

## Round 0.1 · original submission · Major Revisions

First, let me apologize for the delay in making a decision on your manuscript, but I needed to have at least two evaluations from experts in this field. Both reviewers have concerns about aspects of the manuscript in particular about the experimental design, controls and details of the methodology. Please ensure that you respond to all of the concerns in a detailed rebuttal letter. Also, prior to resubmitting your manuscript please ensure that a thororugh and exhuastive revision is made of the entire manuscript with respect to the use of the English language.

**Language Note:** The Academic Editor has identified that the English language must be improved. PeerJ can provide language editing services - please contact us at [email protected] for pricing (be sure to provide your manuscript number and title). Alternatively, you should make your own arrangements to improve the language quality and provide details in your response letter. – PeerJ Staff

·

Basic reporting

The overall structure of the paper is good and figures, data and tables are explanatory and sufficient in general.
Figure 3 could be made more immediate using color or patterns to visually distinguish sediment, pellet and tissue.
Throughout the whole paper English is sometimes incorrect or unclear, I suggest the authors to use a professional editing service to improve it. For example the phrase in lines 51 to 54 is hard to understand and needs to be rephrased.
Line 81 should be rephrased as: ‘Exposure of crab E. sinensis to MPs’
Lines 178 to 180 are not clear: Do the authors mean something like ‘MPs were found in 100% of samples…’?
Lines 201 to 216 are not clear to me.
Other minor grammar mistakes are present throughout the paper and correcting them would improve the readability of the work.
The introduction is complete, informative and supported by sufficient references but informations often look disconnected and simply placed one after the other such as in lines 69 to 76.
Some of the information given should be explained more in depth:
Lines 65 and 219: ‘false ingestion’ is mentioned with no explanation or reference.
Lines 112 to 114: A reference is given but the process used to define impact levels is not explained. The use of the number of visitors per hour as the only criterion to evaluate anthropic impact looks reductive: many other factors could contribute, such as the proximity with urban areas or the presence of discharged waters. It should at least be explained why such a simplistic choice was made.

Experimental design

The research question is clear and the knowledge gaps that the work aims at filling are well explained. The experimental design is simple but sound even though some of the processes are not explained clearly.
Line 116: It should be stated more clearly at what point during the tidal cycle the campaign has been performed.
Lines 123 to 125: It should be made clearer that burrow sediment and feeding pellets are not pooled in a single batch. The phrase ‘mixed together separately’ is unclear.
Line 131: Those are not actual replicas. They may be considered technical replicates or sub-replicates. Explain.
Lines 133 to 136: Providing the quantities of the various solutions that have been used for each process would help with the replicability and consistence.
Line 153: It should be specified how the plastics to be identified were chosen. Were they chosen to be representative of the MPs variability? Were they picked randomly?
A potential flaw of the design is the absence of 'control' samples of sediment unaffected by crabs, the quantification of MPs in the 'normal' sediment would add and interesting insight on the effects of the feeding. That absence should be at least justified.

Validity of the findings

Findings are clear and consequential. Research questions are answered and supported by simple, yet robust, data and analyses. The absence of control samples represents a problem in my opinion but does not invalidate the work overall.

·

Basic reporting

This manuscript titled " Investigating the journey of microplastic from beach sediment to tissue: a case study on burrowing crab Dotilla blanfordi" investigated the abundance of microplastics in sediment, tissues of a burrowing crab, and its feeding pellet. This study is interesting and contributes to our knowledge of microplastic contamination in environmental samples.

Experimental design

My major concern is how can the authors ensure the absence of airborne contamination by microplastics? This is crucial to verify the results. It appears that the authors used replicates along with the digestion step and found no microplastics, but how about other steps? For instance, tissue preparation for digestion should be performed along with blank replicates or under a fume hood to detect or avoid microplastic contamination. Further comments are included below:
1- why sodium chloride was used to isolate microplastics, as some heavy polymers might not be isolated?
2- Line 148: what is the size of the filter used?
3- Line 158: how the samples were covered?

Validity of the findings

Can you find a correlation between microplastic abundance in tissues and in sediment or pellets?

---

## Round 0.2 · Minor Revisions

I have received evaluations from two expert reviewers and their comments can be seen below. Please ensure that you attend to the small list of suggestions. I also strongly encourage that the English useage be improved to prevent further delay of the review process.

·

Basic reporting

The paper improved from a formal point of view event though English is still not great in some parts.

Experimental design

The classification of anthropic impacts has been expanded on a little. I still find it a little simplistic because it ignores important factors such as proximity with urban areas, wastewater/riverine inputs and boat traffic but it's a choice of the authors which is acceptable if consistent between sites.

Validity of the findings

No comment

Additional comments

No comment

·

Basic reporting

please add the results of the current study to the comparison in Tables 2 and 3.

Experimental design

No other comments

Validity of the findings

No other comments

---

## Round 0.3 · Minor Revisions

Thank you for submitting you revised manuscript. I have read the manuscript and agree with the modifications that were made to the manuscript, except that I cannot find any of the corrections to the use of the English language. I cannot find any evidence of this in your tracked changes or clean versions. Please send a revised manuscript that clearly shows where the corrections to the English language have been made (including the title of the manuscript).

---

## Round 0.4 · accepted · Accept

I am satisfied with the changes that have been made to the manuscript. I would suggest that in the title the wording be changed to ...study on the burrowing ... This can be taken care of during the production phase.